# Interplay between Gut Microbiota and NLRP3 Inflammasome in Intracerebral Hemorrhage

**DOI:** 10.3390/nu14245251

**Published:** 2022-12-09

**Authors:** Yuan Zhang, Wanpeng Yu, Christopher Flynn, Wenguang Chang, Lei Zhang, Man Wang, Wanhong Zheng, Peifeng Li

**Affiliations:** 1Institute for Translational Medicine, The Affiliated Hospital of Qingdao University, Qingdao University, Qingdao 266021, China; 2Medical Collage, Qingdao University, Qingdao 266071, China; 3West Virginia School of Osteopathic Medicine, St., Lewisburg, WV 26505, USA; 4Department of Behavioral Medicine and Psychiatry, Rockefeller Neuroscience Institute, West Virginia University, St., Lewisburg, WV 26505, USA

**Keywords:** intracerebral hemorrhage, secondary brain injury, NLRP3 inflammasome, gut microbiota

## Abstract

The pathophysiological process of intracerebral hemorrhage (ICH) is very complex, involving various mechanisms such as apoptosis, oxidative stress and inflammation. As one of the key factors, the inflammatory response is responsible for the pathological process of acute brain injury and is associated with the prognosis of patients. Abnormal or dysregulated inflammatory responses after ICH can aggravate cell damage in the injured brain tissue. The NOD-like receptor family pyrin domain-containing 3 (NLRP3) inflammasome is a multiprotein complex distributed in the cytosol, which can be triggered by multiple signals. The NLRP3 inflammasome is activated after ICH, thus promoting neuroinflammation and aggravating brain edema. In addition, there is evidence that the gut microbiota is crucial in the activation of the NLRP3 inflammasome. The gut microbiota plays a key role in a variety of CNS disorders. Changes in the diversity and species of the gut microbiota affect neuroinflammation through the activation of the NLRP3 inflammasome and the release of inflammatory cytokines. In turn, the gut microbiota composition can be influenced by the activation of the NLRP3 inflammasome. Thereby, the regulation of the microbe–gut–brain axis via the NLRP3 inflammasome may serve as a novel idea for protecting against secondary brain injury (SBI) in ICH patients. Here, we review the recent evidence on the functions of the NLRP3 inflammasome and the gut microbiota in ICH, as well as their interactions, during the pathological process of ICH.

## 1. Introduction

Intracerebral hemorrhage (ICH) refers to the hemorrhage caused by a ruptured blood vessel in the non-traumatic brain parenchyma, which accounts for 20% to 30% of all strokes, with an acute mortality rate of 30% to 40%. The cause is mainly related to cerebrovascular lesions. The pathophysiological process of ICH is very complex, involving various mechanisms such as apoptosis, oxidative stress and inflammation. As one of the key factors, the inflammatory response is responsible for the pathological process of acute brain injury and is associated with the prognosis of patients [1]. Immunity and inflammation play important roles in the process of secondary brain injury (SBI) following ICH. Abnormal or dysregulated inflammatory responses after ICH can aggravate cell damage in the injured brain tissue [2]. It has been reported that the inflammatory responses following ICH begin with the secretion of activated microglia inflammatory factors. After ICH, white blood cells and other components of the peripheral blood enter the brain tissue, activate microglia and promote the production of local inflammatory factors [3]. These inflammatory factors, together with cleavage products resulting from cell death, accelerate the disruption of the blood–brain barrier, damage the brain parenchyma around the lesion and cause the expansion of edema around the hematoma, thereby causing serious damage to the brain tissue.

The NOD-like receptor family pyrin domain-containing 3 (NLRP3) inflammasome is a multiprotein complex distributed in the cytosol, which can be activated by a variety of signals, including bacterial, fungi and viral components, endogenous danger signals and particulates in the environment [4]. Previous studies have shown that the activation of the NLRP3 inflammasome is related to the occurrence and progression of different diseases, including atherosclerosis, metabolic disorders and inflammatory bowel disease [5,6]. As a critical component of innate immunity upon tissue injury, the NLRP3 inflammasome is activated after ICH, thus promoting neuroinflammation and aggravating brain edema. A growing body of evidence suggests that the NLRP3 inflammasome is activated and exhibits a detrimental effect on the brain following ICH [7,8,9].

There is increasing evidence that the NLRP3 inflammasome may be at the crossroads of the brain–gut–microbe connection. The gut microbiota is crucial in the activation of the NLRP3 inflammasome (an important mediator of microbiome-induced inflammatory responses) [10,11,12]. The gut microbiota is a very large group of microorganisms that live in the human gut. The gut microbiota has a direct functional relationship with the central nervous system (CNS), in the so-called “gut–brain axis”. The gut microbiota plays a key role in a variety of CNS disorders, including ischemic stroke, Parkinson’s disease and Alzheimer’s disease [13,14,15]. The gut microbiota can affect the outcome of acute brain injury by modulating the immune system [16]. It has been reported that the dysregulation of the gut microbiota triggers neuroinflammation and increases disease severity in animal ICH models [17,18]. Changes in the gut microbiota influence neuroinflammation by activating the NLRP3 inflammasome and promoting the release of inflammatory cytokines. In turn, the gut microbiota composition can be influenced by the activation of the NLRP3 inflammasome [19]. Thereby, the regulation of the microbe–gut–brain axis via the NLRP3 inflammasome may serve as a novel idea for the treatment of ICH patients. Here, we review the recent evidence on the functions of the NLRP3 inflammasome and the gut microbiota in ICH, as well as their interactions, during the pathological process of ICH.

## 2. Functions of the NLRP3 Inflammasome in ICH

### 2.1. Activation of the NLRP3 Inflammasome

The NLRP3 inflammasome consists of the effector protein pro-caspase-1, the adapter protein apoptosis-associated speck-like protein (ASC) and the sensor protein NLRP3 and orchestrates innate immune responses against cell stress and infection by regulating the caspase-1-dependent pathway and releasing proinflammatory cytokines such as interleukin-1β (IL-1β) and IL-18 [20].

The sensor protein NLRP3 consists of a carboxy-terminal leucine-rich repeat (LRR) domain, a NACHT domain and a pyrin domain (PYD). Upon binding to the corresponding ligands, thee LRR domain regulates the function of LRRs. The activation of the NLRP3 inflammasome involves two stages, i.e., the priming and activation stages (Figure 1). The priming stage is induced by the recognition of two molecular patterns, i.e., the damage-associated molecular patterns (DAMPs) and the pathogen-associated molecular patterns (PAMPs) [21]. This can activate the NF-κB signaling pathway and promote the expression of precursor proteins, including pro-IL-1β, pro-IL-18 and NLRP3. The activation stage is triggered by multiple stimuli that exist during metabolic imbalance, infection or tissue injury. When dangerous signals are recognized, the NLRP3 protein structure changes and exposes its PYD domain, which will bind to the PYD of ASC by forming a PYD–PYD interaction. Subsequently, ASC recruits the cysteine protease pro-caspase-1 to assemble the inflammasome complex via interacting with the caspase recruitment domain (CARD). Pro-caspase-1 is activated by self-cleavage to form active caspase-1. Then, caspase-1 dissociates gasdermin D (GSDMD) to release its *N*-terminal domain, which in turn binds to phosphatidylinositol phosphates and phosphatidylserine in the cytomembrane to generate pores, thereby inducing a lytic form of cell death, known as “pyroptosis”. In addition, caspase-1 can induce the transformation of IL-1β and IL-18 precursors into mature IL-1β and IL-18 and eventually aggravate the inflammatory responses and related complications.

### 2.2. Modulation of NLRP3 Inflammasome Activity as a Therapeutic Strategy for SBI after ICH

The NLRP3 inflammasome is a multimolecular complex in the cytoplasm that mediates caspase-1 processing and proinflammatory cytokine maturation, including IL-1β and IL-18. Various risk factors such as Ca^2+^ mobilization, Na^+^ influx, K^+^ efflux, chloride efflux, mitochondrial dysfunction, oxidative stress and lysosomal damage are involved in the activation of the NLRP3 inflammasome to mediate neuroinflammatory responses after ICH [9,22]. Activating the NLRP3 inflammasome generates high levels of inflammatory cytokines, triggers an inflammatory response and recruits other immune cells to clear DAMPs after hemorrhage. However, overactivation of the NLRP3 inflammasome can result in persistent neuroinflammation and brain injury after ICH. Thus, assessing the role of the NLRP3 inflammasome in the processes associated with ICH may provide new strategies for ICH therapy.

As a crucial component of innate immunity upon tissue injury, the NLRP3 inflammasome is activated after ICH, thereby promoting neuroinflammation and aggravating brain edema [8]. It has been reported that the expression of NLRP3 is gradually upregulated in the perihematoma tissue within 1–5 days after ICH, and the NLRP3 inflammasome is responsible for the complement-induced neuroinflammation, which eventually leads to abnormal neurological functions [9]. Activating the NLRP3 inflammasome can promote neuroinflammation via caspase-1 processing and IL-1β generation following ICH. Nevertheless, ICH-induced NLRP3 inflammasome activation can promote neutrophil infiltration, trigger the inflammatory response, impair neurological functions and aggravate brain edema after ICH [9].

In recent years, numerous studies have been conducted around the functions of the NLRP3 inflammasome in ICH. Brain injury induced by inflammation after ICH can be alleviated by directly or indirectly inhibiting NLRP3 inflammasome activation. For example, histone deacetylase 10 (HDAC10) downregulates protein tyrosine phosphatase nonreceptor type 22 (PTPN22) expression by binding to and deacetylating the PTPN22 promoter, which inhibits NLRP3 inflammasome activation and alleviates inflammation after ICH in rats. PTPN22 is a protein tyrosine phosphatase involved in the cellular immune response and inflammation and is related to various autoimmune diseases. PTPN22 binds and dephosphorylates NLRP3 following proinflammatory injury, thus promoting NLRP3 activation and IL-1β secretion. Interfering with PTPN22 can reduce the expression of IL-1β and IL-18 by inhibiting the activation of the NLRP3 inflammasome to reduce inflammatory responses, thereby improving neurological dysfunction and reducing cerebral edema in ICH rats [23]. Mammalian sterile-20-like kinase 4 (MST4) is a member of the glucokinase (GCK) subfamily, which directly phosphorylates TRAF6 to suppress inflammation. MST4 phosphorylates and activates TRAF6, which further activates the NLRP3 inflammasome by regulating the TLR/IL-1R signaling pathway [7]. Overexpression of MST4 negatively regulates the NLRP3 inflammasome and reduces the expression of tumor necrosis factor-α (TNF-α) and IL-1β, indicating that NLRP3 and MST4 may be potential therapeutic targets for neuroinflammation after ICH [7].

Mitochondrial dysfunction plays an essential role in the activation of the NLRP3 inflammasome [24]. Mitophagy is involved in the maintenance of mitochondrial homeostasis via selective degradation of damaged mitochondria, which prevents inflammation by inhibiting the NLRP3 inflammasome pathway [25]. Therefore, modulating mitophagy to inhibit NLRP3 inflammasome activation can become a therapeutic approach for alleviating secondary brain injury (SBI) after ICH. Chen et al. revealed that the activation of the Nrf2/Optineurin (OPTN) pathway can mediate mitophagy to alleviate SBI by suppressing the NLRP3 inflammasome following ICH. OPTN is a multifunctional ubiquitin-binding autophagy receptor that interacts with Nrf2 to mediate mitophagy and eliminate damaged mitochondria following ICH. Suppressing Nrf2/OPTN could enhance NLRP3 inflammasome activation, downregulate mitophagy levels and increase BBB disruption and brain edema, with more severe neurological deficits after ICH [26]. FUN14 domain-containing 1 (FUNDC1) is a mitophagy receptor that is overexpressed after ICH. FUN14 can suppress the NLRP3 inflammasome via regulation of mitophagy, thus alleviating ICH-induced injury. However, silencing FUNDC1 promotes NLRP3-mediated inflammation, thereby suppressing mitophagy and exacerbating ICH injury [27]. Thus, it can be seen that modulating mitophagy to inhibit NLRP3 inflammasome activation is an important therapeutic strategy for alleviating ICH injury.

Many drugs can improve brain dysfunction and protect against SBI after ICH by acting NLRP3 inflammasome pathways (Table 1). For example, pioglitazone, edaravone and adiponectin significantly reduced brain edema and attenuated neurological deficits after ICH, as well as decreased the expression of IL-1β, IL-18, caspase-1 and NF-κB through suppressing the expression of NLRP3 [8,28,29]. Glibenclamide markedly reduced the neurological deficit and brain edema after ICH by decreasing the expression of ASC and caspase-1 and suppressing the activation of the NLRP3 inflammasome to maintain BBB integrity [30]. Memantine reduces ONOO- production by inhibiting neuronal nitric oxide synthase (nNOS) phosphorylation at ser1412, which further inhibits MMP-9 expression and NLRP3 inflammasome activation, protecting the blood–brain barrier integrity and alleviating neurological deficits in ICH rats [31]. Atorvastatin can protect the neurological function and reduce neuroinflammation and neuronal apoptosis by reducing the expression of TNF-α, IL-6 and IL-1β in ICH model mice. Furthermore, atorvastatin can decrease the expression of NLRP3 and cleaved caspase-1 and reverse the increase in toll-like receptor 4 (TLR4) and myeloid differentiation factor 88 (MyD88), indicating that atorvastatin suppresses NLRP3 inflammasome activation in glial cells of ICH model mice through inhibiting MyD88- and TLR4-associated pathways [32]. Some traditional Chinese medicine such as isoliquiritigenin, silymarin, baicalein and cordycepin can also exert antioxidant and anti-inflammatory roles by significantly inhibiting the activation of the Nrf2 dependent-NF-κB pathway and NLRP3 inflammasome, which in turn mitigates brain edema and improves neurological deficits after ICH [33,34,35,36,37]. This evidence indicates that the suppression of NLRP3 inflammasome activation might be a therapeutic target for ICH recovery.

## 3. The Function of the Gut Microbiome in ICH

### 3.1. Gut Microbiota Dysbiosis Is Induced after ICH

As the most important and complex micro-ecosystem in the human body, the intestinal micro-ecosystem includes three phyla, *Firmicutes*, *Bacteroidetes* and *Actinobacteria* and about 10 trillion and more than 3500 kinds of bacteria. Among them, *Bacteroidetes*, *Bifidobacterium*, *Enterobacter*, *Lactobacillus*, *Enterococcus*, *Clostridium* and *Staphylococcus* are the main anaerobic bacteria. The diversity, complexity and dynamic changes of the gut microbial composition play a crucial role in human health. It has been reported that the gut microbes can regulate various pathophysiological changes in the human body. The composition and diversity of the gut microbiome may be influenced by gut motility, transit, barrier integrity, and secretion of various factors, which are modulated by enteric nervous system activity and mediated by CNS inputs. Under normal physiological conditions, the intestinal microflora maintains a relatively stable state to ensure the health of the body. However, changes in the internal and external environment of the body can lead to gut microbiota dysbiosis, which affects the host’s metabolism and immune response.

In patients with acute CNS injury such as stroke or craniocerebral trauma, the intestinal mucosa is mostly the first organ affected under stress. Once the intestinal mucosa is damaged, the number and structure of the intestinal flora will change immediately. Previous studies found that the gut microbiota changed significantly through the brain–gut axis after ICH [38] (Table 2). After stroke, activated microglia and injured brain tissue release DAMPs and cytokines, leading to endothelial cell activation and the expression of chemokines and adhesion molecules, which further promote the recruitment of immune and inflammatory cells from the bloodstream to the damaged site in the brain [39]. At the same time, the activation of the vagus nerve as well as the release of cytokines and DAMPs improve gut dysbiosis, dysmotility and permeability. The release of cytokines leads to a decrease in the diversity of the gut microbiota, which in turn triggers an immune response and intestinal inflammation and alters the immune homeostasis. For example, Yu et al. reported that increased levels of regulatory T lymphocytes (Tregs), CD4+ T and CD8+ T cells in the hemorrhagic hemisphere indicated that T cells infiltrated into the perihematomal region after ICH. Furthermore, the levels of IL-1β, TNF-α and inducible nitric oxide synthase (iNOS) were significantly increased in the brain following ICH. With the activation of immune cells and the secretion of cytokines, members of *Helicobacteraceae*, *Nocardiaceae*, *Veillonellaceae*, *Akkermansiaceae* and *Bacteroidaceae* families were significantly upregulated, while *Barnesiellaceae*, *Firmicutes*, *Moraxellaceae* and *Bacteriidales* decreased, along with impaired gastrointestinal function and increased gut permeability [17]. Another cohort study of patients with hypertensive intracerebral hemorrhage (HICH) showed that the phyla *Verrucomicrobia*, *Proteobacteria* and *Firmicutes* were markedly altered in patients with HICH. The genera *Blautia*, *Bacteroides*, *Subdoligranulum*, *Faecalibacterium*, *Bifidobacterium*, *Romboutsia* and *Agathobacter* were significantly decreased, whereas *Akkermansia*, *Escherichia-Shigella*, *Lachnoclostridium* and *Lactobacillus* increased significantly after HICH [40]. Xiong et al. found that four phyla and six orders of bacterial species, including *Coriobacteriales* (*phylum Actinobacteriota*), *Actinomycetales* (*phylum Actinobacteriota*), *RF39* (*phylum Firmicutes*), *Gastranaerophilales* (*phylum Cyanobacteria*), *Verrucomivrobiales* (*phylum Verrucomicrobiota*) and *TANB77* (*phylum Firmicutes*), were significantly changed in patients with intraparenchymal hemorrhage (IPH). Among them, *Verrucomivrobiales* (*phylum Verrucomicrobiota*) was identified to play an important role in distinguishing IPH patients [41]. Additionally, it was found that when the gut microbiota of rats with hemorrhagic transformation was introduced in the intestinal tract of normal rats, the susceptibility of the latter to hemorrhagic transformation increased [42]. In addition, Luo et al. reported that the relative abundances of *Enterococcus*, *Parabacteroides*, *Lachnoclostridium*, *Acidaminococcus* and *Streptococcus* were higher in ICH patients than in healthy controls, while the relative abundances of *Prevotella* and *Faecalibacterium* were lower in ICH patients than in healthy controls. Changes in the composition of the gut microbiota are closely related to changes in the levels of inflammatory factors, for example, the levels of GM-CSF, IL-12p70, IL-15, IL-1RA, IL-9, IL-23 and TNF-a increased, and the levels of IL-10 decreased in ICH patients. This study indicated that an increase in *Enterococcus* and a decrease in *Prevotella* increased the risk of ICH [43]. Thus, the diversity of the gut microbiota decreased after ICH, as some phyla and genera increased in abundance, and some decreased. Changes in the composition of the microbiota are regulated by the brain–gut–microbiota axis, which can exacerbate the pathological process of ICH.

### 3.2. Gut Dysbiosis Exacerbates the Neuroinflammatory Response of ICH

The gut microbiota stability plays a vital role in regulating brain development and maintaining host–microbe homeostasis. The gut microbiota can regulate a series of neurotrophic factors or proteins associated with brain development and plasticity, including brain-derived neurotrophic factor (BDNF), synaptophysin (SYP) and postsynaptic dense zone proteins (PSD-95 and SynGAP). An altered gut microbiota largely affects the host’s neurological functions and behaviors, while modulating the repair process of post-neurological trauma. Many experimental and clinical studies demonstrated the importance of the gut microbiota in the pathogenesis and course of ischemic stroke and suggested that the gut microbiota may influence the outcome of brain injury by regulating the CNS antigen-specific immune responses [44].

The gut microbiota can affect the brain by regulating the brain–gut–microbiota axis, and the dysregulation of its composition may be a therapeutic target for ICH. Thus, the gut microbiota instability is a vital factor in the regulation of neuroinflammation after ICH, which alters immune homeostasis, induces pro-inflammatory responses and leads to deteriorated outcomes. The gut microbiota dysbiosis exacerbates the inflammatory response induced by ICH. Numerous T cells migrate from the intestine to the peri-hematoma regions, exacerbating neuroinflammation after ICH. For example, the activation of intestinal γδ T cells can exacerbate brain injury by attracting monocytes and neutrophils at the injury site and promoting the secretion of proinflammatory cytokines. However, in the late stages of ICH, dendritic cells in the mesenteric lymph nodes promote the migration of Treg cells to the intestine, thereby inhibiting the differentiation of IL-17-producing γδ T cells and attenuating the migration of γδ T cells from the gut to the brain and ultimately protecting against cerebral ischemia/reperfusion injury [45].

In addition, gut bacteria generate neuroactive compounds and mediate neuronal functions, thus affecting the behavior after ICH. Previous research demonstrated that the gut microbiota affects the host homeostasis through bile acid metabolism, amino acid metabolism and metabolic pathways such as those involving SCFAs. The gut microbiota modulates the balance between disease and health because it influences many processes of metabolism: from producing metabolic precursors for neurotransmitter and hormone metabolism to directly producing active metabolites, such as SCFAs, trimethylamine oxide, amino acids, vitamins and bile acids, which may be responsible for modulating the host’s physiological functions [46,47]. For instance, trimethylamine oxide enhances platelet reactivity and thrombotic potential, which may be a potential causative factor of ISH. Furthermore, SCFAs may be important factors in suppressing neuroinflammation. It has been reported that acetate can modulate neurotransmitters, such as γ-aminobutyric acid (GABA), glutamine, glutamate and neurotrophic factors (NTFs) [48]. A recent study showed that SCFAs prevented fructose-induced gut dysbiosis in mice, which in turn alleviated dysbiosis-induced hippocampal inflammatory responses and neuronal loss [49]. SCFAs were demonstrated to regulate the maturation and function of microglia in the brain, and thus, gut microbiota-derived SCFAs may protect against neuroinflammation after brain damage [50,51]. Studies have shown that the levels of acetic acid are remarkably lower in stroke patients, whereas those of valeric acid are markedly higher. However, the concentrations of total organic acid were reduced in stroke patients. Stroke patients had lower levels of SCFAs-producing bacteria and SCFAs. For example, during middle cerebral artery occlusion, mice had lower levels of SCFAs, especially butyrate and valeric acid [42]. The capacity of carbohydrate and amino acid metabolism is increased in the gut bacteria of IPH patients. The metabolism of the gut microbiota is accelerated, as a large amount of lipids, amino acids and carbohydrates is needed to generate energy and produce beneficial neuroactive compounds, so as to alleviate brain injury after cerebral hemorrhage [41].

Intestinal damage caused by ICH induces gut dysbiosis, leading to the aggravation of the inflammatory responses, while improving the gut microbiota through drug regulation, metabolite regulation or gut microbiota transplantation can alleviate neuroinflammation after ICH. For example, metformin can regulate the gut microbiota composition to suppress neuroinflammation and promote neurofunctional recovery in an ICH mouse model. Fecal microbiota transplantation from metformin-treated mice could reduce neuroinflammation by modulating the microglial/macrophage phenotype after ICH [18]. The transplantation of the normal microbiota into ICH mice alleviated the gut barrier damage, ameliorated neurological deficits and suppressed neuroinflammation after ICH [17]. Hence, the gut microbiota can play an essential role in attenuating ICH-induced brain damage via neuroinflammation pathways.

## 4. Interplay between the Gut Microbiota and the NLRP3 Inflammasome during ICH

The NLRP3 inflammasome is considered a major player in neuroinflammation and has been shown to be a key factor in promoting proinflammatory cytokine release and subsequent inflammatory responses. Suppression of the NLRP3 inflammasome attenuates secondary brain injury caused by ICH. The NLRP3 inflammasome may be at the crossroads of the brain–gut–microbe connection. Hence, modulating the brain–gut–microbe axis via the NLRP3 inflammasome is a novel therapeutic strategy for patients with ICH.

### 4.1. NLRP3 Inflammasome Signaling Modulates the Gut Microbiota Composition

Evidence shows that the gut microbiota composition is influenced by the activation of the NLRP3 inflammasome [52]. A previous study demonstrated that wild-type (WT) mice and NLRP3 knockout (KO) mice differed in gut microbiota load and species [53]. Moreover, *Firmicutes* were obviously increased, while *Bacteroidetes* were dramatically reduced in NLRP3 KO mice [53]. Furthermore, at the family level, *Prevotellaceae*, *Ruminococcaceae* and *Lachnospiraceae* displayed a significant increase in NLRP3 KO mice. Another report showed that the ratio of *Firmicutes* to *Bacteroidetes* was significantly decreased, and the abundance of *Prevotellacea* was increased in caspase-1 KO mice, indicating that NLRP3 inflammasome signaling plays an important role in the inflammatory response by regulating the composition of enteric bacteria [54]. However, suppressing the NLRP3 inflammatory pathway alters the structure and diversity of the gut microbiota and its metabolites and, most importantly, inhibits the inflammatory responses. Another study showed that the suppression of the NLRP3 inflammatory pathway by the inhibitor NU9056 can ameliorate the dysregulated function of microglia and the inflammatory responses induced by lipopolysaccharide [55]. Treatment with NU9056 increased *Verrucomicrobia* at the phylum level and *Akkermansia* at the genus level, compared to LPS-treated mice. Furthermore, NU9056 could reverse the decreased concentrations of acetate, propionic acid and butyrate induced by IPS [55]. The results suggest that the suppression of the NLRP3 inflammatory pathway changes the diversity and structure of the gut microbiota and its derived metabolites. In particular, when the diversity and structure of the gut microbiota change, their metabolites will also be altered significantly. When the bacterial species producing SCFAs decline, the SCFAs produced by metabolism also decrease. Changes in metabolites and their content may affect the function of the organism and brain through the blood circulation.

Thus, the inhibition of the NLRP3 inflammasome can modulate the gut microbiota composition after ICH. ICH can induce gut microbiota dysbiosis, and inhibition of the NLRP3 inflammasome can relieve brain edema and attenuate corticospinal tract injury by changing the bacterial community structure in ICH patients. The effect of NLRP3 inflammasome inhibition may be promoted by inducing an enrichment of beneficial bacteria including *Bacteroides* and *Bifidobacterium* and reducing the number of harmful bacteria including *Helicobacter*, *Enterobacter* and *Desulfovibrionaceae* [56]. Therefore, treatment with MCC950 (an NLRP3 inflammasome inhibitor) can facilitate the shift of the gut microbiota in a favorable direction after ICH.

### 4.2. Effects of the Gut Microbiota on NLRP3 Inflammatory Responses

The gut microbiota is involved in the activation of the NLRP3 inflammasome that is an important regulator of microbiome-induced inflammatory responses. The gut microbiota can trigger inflammatory responses and affect disease outcomes by activating the NLRP3 inflammasome. Changes in the gut microbiota can affect neuroinflammation by activating the NLRP3 inflammasome and promoting the secretion of inflammatory cytokines [57,58]. The interplay between the gut microbiota and the NLRP3 inflammasome in neuroinflammation can be explained by the biological pathways influenced by the gut microbiota–inflammatory response–brain axis. Under pathological conditions such as ICH, the number of beneficial bacteria decreased in the intestinal tract, while that of unfavorable bacteria increased. The increased number of adverse bacteria in the intestinal tract, accompanied by the increase in harmful microbial products such as endotoxin, can lead to a gut barrier dysfunction, which induces the secretion of inflammatory cytokines from intestinal parietal cells, gut bacteria-derived pathogens and toxins (e.g., LPS) into the systemic circulation and activates peripheral NLRP3-associated inflammation. There is increasing evidence that an unfavorable gut microbiota is associated with the overexpression of NLRP3 in the gut, thereby activating the peripheral inflammasome and exacerbating neuroinflammation. NLRP3 can recognize PAMPs expressed by the gut microbiota and potentially cytotoxic mediators in the gut, leading to increased levels of gut inflammasome components. The microbial product LPS has also been shown to activate the NLRP3 inflammasome. However, depleting the gut microbiota by an antibiotic cocktail can influence inflammasome signaling in the brain, blood and intestine [59].

The interaction of the NLRP3 inflammasome with gut microbiota dysbiosis has been reported in several neurodegenerative diseases. For example, Shen et al. showed that fecal transplantation from Alzheimer’s disease patients in APP/PS1 transgenic mice could induce the activation of the gut NLRP3 inflammasome and the secretion of proinflammatory cytokines, thus aggravating neuroinflammation and cognitive deficits [11]. Shukla and colleagues found that the intestinal barrier function was impaired and the activation of the NLRP3 inflammasome was increased in the gut of 5xFAD mice, which was accompanied by an increase in *Bacteroidetes* and a loss of beneficial bacteria (e.g., *Bifidobacterium*) [52]. The elevation of the NLRP3 inflammasome in the intestine led to peripheral inflammasome activation through the release of inflammatory cytokines, triggering NLRP3-mediated neuroinflammation in the brain of 5xFAD mice [52]. Another study showed that the induction of colitis in germ-free (GF) mice did not induce IL-1β production in colonic lamina propria (LP) cells. In vitro stimulation of bone marrow-derived macrophages (BMDMs) with fecal contents from specific-pathogen-free (SPF), but not GF mice, could result in the production of IL-1β [60]. These findings indicate that the activation of the NLRP3 inflammasome needs the participation of the gut microbiota.

In addition, a series of metabolites produced by the gut microbiota significantly affect the activity of the NLRP3 inflammasome. SCFAs, as the major metabolites produced by the bacterial fermentation of dietary fiber in the gastrointestinal tract, have an important regulatory role in the CNS [61]. Previous evidence demonstrated that SCFAs act as energy substances and HDAC inhibitors to repair the dysregulated intestinal barrier and suppress the NLRP3 inflammasome [62]. Feng et al. also showed that SCFAs inhibited the LPS-induced overexpression of ASC, NLRP3, IL-1β, cleaved-IL-1β, IL-18, caspase-1 and cleaved-caspase-1 in Caco-2 cells, implying that SCFAs can act as HDAC inhibitors to regulate the NLRP3 inflammasome pathway [62]. However, other studies also indicated that SCFAs, as metabolites, could activate the gut NLRP3 inflammasome and subsequently secrete inflammatory factors that reach the central system through the bloodstream and induce neuroinflammation [63]. This may be due to the fact that under different pathological conditions, different gut microbiota and their metabolites lead to different downstream signaling pathways, resulting in different regulatory effects. Taken together, SCFAs can affect the central system by modulating the NLRP3 inflammasome pathway.

## 5. Conclusions

Increasing evidence has suggested that the NLRP3 inflammasome and the gut microbiota are pivotal factors that affect the pathological process of ICH. They have also been identified as novel therapeutic targets for ICH. The gut microbiota and its metabolites may play a crucial role in the activation of the NLRP3 inflammasome (Figure 2). Although the respective roles of the NLRP3 inflammasome and the gut microbiota in the pathological process of ICH have been gradually elucidated, the impact of their interaction in the pathophysiology of ICH has rarely been studied in depth. Therefore, elucidating the interaction between the NLRP3 inflammasome and intestinal microecology in ICH is important for a better understanding of the pathological mechanisms of ICH.

The NLRP3 inflammasome is activated after intracerebral hemorrhage, thus promoting neuroinflammation and aggravating brain edema. ICH-induced NLRP3 inflammasome activation can promote neutrophil infiltration, trigger the inflammatory response, impair neurological functions and aggravate brain edema after ICH. Brain injury induced by inflammation after ICH can be alleviated by directly or indirectly inhibiting NLRP3 inflammasome activation. In addition, the diversity of the gut microbiota is reduced after ICH, as some phyla and genera increased in abundance and some decreased. Changes in the composition of the microbiota are regulated by the brain–gut–microbiota axis and exacerbate the pathological process of ICH. Furthermore, the gut bacteria generate neuroactive compounds and regulate neuronal functions, which can affect the behavior after ICH.

The gut microbiota is crucial for the activation of the NLRP3 inflammasome that is involved in microbiota-induced inflammatory responses. ICH triggers NLRP3 inflammasome activation, changing the gut microbiota diversity and its metabolites. In turn, the altered gut microbiota can affect neuroinflammation by activating the NLRP3 inflammasome and secreting inflammatory cytokines. Nevertheless, further in-depth studies of the NLRP3 inflammasome and the gut microbiota metabolites are needed to assist clinical interventions to treat ICH. Therapeutic measures, including dietary interventions, the use of probiotics, fecal microbial transplantation and the administration of small molecule compounds to restore the intestinal microecological homeostasis, are favorable for the prevention and treatment of ICH. Therefore, it is of great importance to clarify the regulatory mechanisms of the interaction between the NLRP3 inflammasome and intestinal microecology in patients with ICH.

## Figures and Tables

**Figure 1 nutrients-14-05251-f001:**
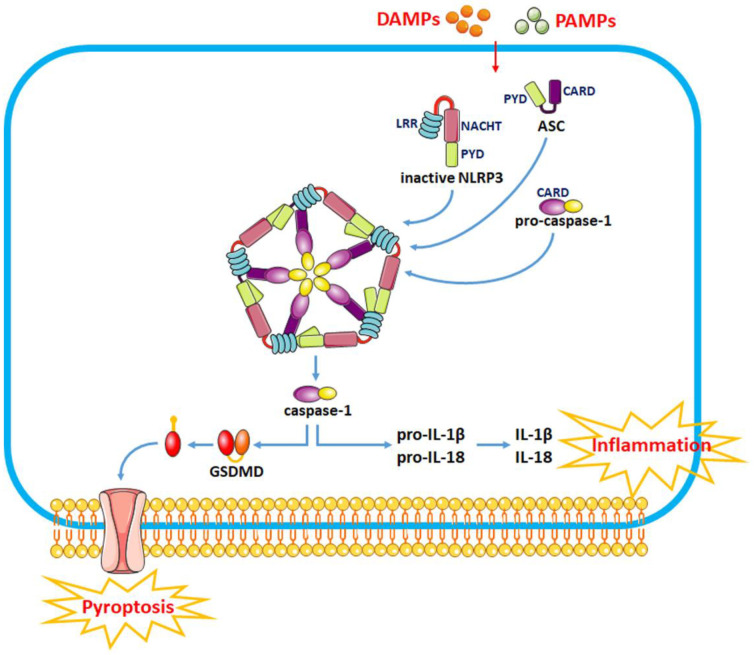
The recognition of DAMPs and PAMPs induces a change in the NLRP3 protein structure and the exposure of its PYD domain to bind to the PYD of ASC, thus forming a PYD–PYD interaction. Then, ASC recruits the cysteine protease pro-caspase-1 to assemble the inflammasome complex via interacting with CARD. Pro-caspase-1 is activated by self-cleavage to form active caspase-1. Activated caspase-1 dissociates GSDMD and releases its *N*-terminal domain, which interacts with the cell membrane to produce pores and induce “pyroptosis”. In addition, caspase-1 can also induce the transformation of IL-1β and IL-18 precursors into mature IL-1β and IL-18 to induce inflammatory responses.

**Figure 2 nutrients-14-05251-f002:**
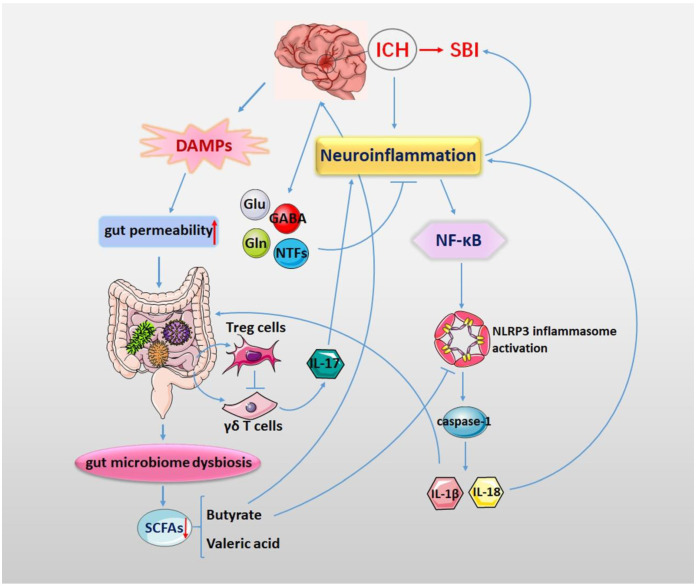
Neuroinflammation plays an important role in the process of SBI after ICH. ICH triggers NLRP3 inflammasome activation. The activation of the NF-κB signaling pathway promotes NLRP3 inflammasome activation and further promotes the release of IL-1β and IL-18 through caspase-1. The release of IL-1β and IL-18 further aggravates the neuroinflammatory response of ICH. The gut microbiota is important in the activation of the NLRP3 inflammasome. After ICH, the release of DAMPs and cytokines as well as the activation of the vagus nerve induce gut dysmotility, gut dysbiosis and increased gut permeability. Gut microbiota dysbiosis will lead to changes in microbial metabolites, especially a reduction in SCFAs. SCFAs act as energy substances and can suppress the NLRP3 inflammasome and repair the dysfunction of the intestinal barrier. ICH triggers NLRP3 inflammasome activation, changing the gut microbiota diversity and its metabolites. In turn, changes in the gut microbiota can influence neuroinflammation by activating the gut NLRP3 inflammasome and releasing inflammatory cytokines.

**Table 1 nutrients-14-05251-t001:** Potential NLRP3 inflammasome inhibitors for protecting against SBI after ICH.

Drugs	Models	Efficacy	References
Pioglitazone	blood-induced mouse ICH model	brain edema↓, lactate↑	[28]
Edaravone	autologous blood-induced rat ICH model	IL-1β↓, caspase-1↓, NF-κB↓, brain edema↓, neurological deficits↓	[29]
Adiponectin	autologous blood-induced rat ICH model	IL-1β↓, IL-18↓, brain edema↓, neurological deficits↓	[8]
Glibenclamide	autologous blood-induced mouse ICH model	IL-1β↓, IL-18↓, IL-6↓, TNF-α↓, brain edema↓, disrupted BBB↓, neurological deficits↓	[30]
Memantine	collagenase-induced rat ICH model	IL-1β↓, disrupted BBB↓, neurological deficits↓	[31]
Atorvastatin	collagenase-induced mouse ICH model	IL-1β↓, IL-6↓, TNF-α↓, brain edema↓, neurological deficits↓	[32]
Isoliquiritigenin	collagenase IV-induced rat ICH model	NF-κB↓, IL-1β↓, brain edema↓, disrupted BBB↓, neurological deficits↓	[36]
Silymarin	collagenase II-induced mouse ICH model	NF-κB↓, caspase-1↓, IL-1β↓	[34]
Baicalein	collagenase VII-induced rat ICH model	ROS↓, SOD↑, GSH-Px↑, ASC↓, caspase-1↓	[35]
Cordycepin	autologous blood-induced mouse ICH model	IL-1β↓, IL-18↓, brain edema↓, neurological deficits↓	[37]

**Table 2 nutrients-14-05251-t002:** Changes in the gut microbiota after intracerebral hemorrhage.

Disease	Models	Gut Microbiota Analysis	Composition of Gut Microbiota	Functions	References
Intracerebral hemorrhage (ICH)	collagenase VII-induced mouse ICH model	16S rRNA sequence	members of *Nocardiaceae*, *Helicobacteraceae*, *Veillonellaceae*, *Bacteroidaceae*, and *Akkermansiaceae*↑ members of *Firmicutes*, *Barnesiellaceae*, *Bacteriidales*, and *Moraxellaceae* ↓	gastrointestinal function impaired, and gut permeability increased	[17]
Hypertensive intracerebral hemorrhage (HICH)	hemoglobin-induced rat HICH model/HICH patients	16S rRNA sequence	the genera *Escherichia-Shigella*, *Akkermansia*, *Lactobacillus* and *Lachnoclostridium*↑ the genera *Bacteroides*, *Blautia*, *Faecalibacterium*, *Subdoligranulum*, *Bifidobacterium*, *Agathobacter* and *Romboutsia* ↓	IL-1β↑, TNF-α↑, IL-10↓*Bacteroides* levels were negatively correlated with IL-1β and TNF-α levels; *Blautia* levels were negatively correlated with IL-1β and TNF-α levels and positively correlated with IL-10 level; *Akkermansia* levels were negatively correlated with 3,7-dimethyluric acid and 7-methylxanthine levels	[40]
Cerebral intraparenchymal hemorrhage (IPH)	IPH patients	metagenomic shotgun sequencing	*Actinomycetales* (*phylum Actinobacteriota*), *Coriobacteriales* (*phylum Actinobacteriota*), *Gastranaerophilales* (*phylum Cyanobacteria*), *RF39* (*phylum Firmicutes*), *TANB77* (*phylum Firmicutes*) and *Verrucomivrobiales* (*phylum Verrucomicrobiota*) changed; *Verrucomivrobiales* (*phylum Verrucomicrobiota*)↑	capacity of energy production and conversion↑, lipid metabolism capacity↓	[41]
Intracerebral hemorrhage (ICH)	ICH patients	16S rRNA sequence	the abundances of *Enterococcus*, *Parabacteroides*, *Lachnoclostridium*, *Acidaminococcus* and *Streptococcus*↑ the abundances of *Prevotella* and *Faecalibacterium*↓	the levels of GM-CSF, IL-12p70, IL-15, IL-1RA, IL-9, IL-23 and TNF-a increased, and the levels of IL-10 decreased	[43]

## Data Availability

Not applicable.

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
