# Peer review of "Interplay between Gut Microbiota and NLRP3 Inflammasome in Intracerebral Hemorrhage"

_nutrients, 2022, doi:10.3390/nu14245251_

Round 1

Reviewer 1 Report

The manuscript by Zhang and colleagues entitled: “Interplay between gut microbiota and NLRP3 inflam-masome in intracerebral haemorrhage” is a well-written review focused on the interaction between NLRP3 inflammosome and intestinal microbiota in intracerebral haemorrhage (ICH). Although the role of NLRP3 inflammosome on (ICH) has been previously described, similarly as the role of gut microbiota dysbiosis on ICH, the interaction among both processes in the context of ICH.

The role of NLRP3 and gut microbiota are described in this review as causative factors of the secondary brain injury (SBI) after the ICH, which aggravates the inflammatory state and worsen the consequences of such event. However, sometimes the authors mention the role of modulators of both NLRP3 and microbiota to prevent the ICH, which is not right based on the data/studies presented. In the same line, I have some concerns that should be improved.

-Section 1: Although references 7 and 8 are appropriate, I suggest including some older to show (as mentioned in the text) that there is an important body of evidence concerning this issue (for instance, cite the study from Ma et al., Ann Neurol 2014).

-Section 2, subheading 2.1: in the second paraghaph, please correct thee. In the same section, I strongly suggest to include a figure showing the structure and activation process of NLRP3, since this is one of the main points of the review.

-Section 2, subheading 2.2.: title. Do the authors believe that this section is focused on therapeutic targets for ICH or for secondary brain injury produced after ICH? Please modify according the statement (third paragraph) that reads as follows: Brain injury induced by inflammation after ICH can be alleviated by directly or indirectly inhibiting NLRP3 inflammasome activation.

-In the same line, last paragraph of this section: the authors cite that: “Many drugs can improve brain dysfunction and prevent ICH injury by acting on NLRP3 inflammasome pathways”. However, the point is not to prevent the development of ICH but to improve the secondary brain injury… Please modify that sentence and also the heading of the table according to this.

-Fourth paragraph: authors use SBI abbreviation for "surgical brain injury" while it was previosly used for secondary brain injury. Please correct.

-Table 1: It should be very interesting and will strengthen the review if studies focused on humans are also included. Atorvastatin study (ref. 31) has also used human brains… It should be highlighted.

-Table 2: only three studies have been included. Please include more studies. I also strongly suggest changing the heading of the table. It should be better if it reads as: “The changes in gut microbiota after (instead of in) intracerebral haemorrhage”.

-Page 7: when talking about SCFAs (ref. 44) specify that is acetate the fatty acid responsible for the actions mentioned.

-Section 4  page 8: Please include the SBI abbreviation in the first paragraph

-Figure 1 legend: the neuroinflammation plays (“s” added).

-Last paragraph of the manuscript: change “to change” by “by changing”.

Author Response

   Thank you for  the reviewer's comments concerning our manuscript. Those comments are all valuable and very helpful for revising and improving our paper. We did our best to revise the manuscript point-by-point and hope that the correction will meet with approval.  
   We hope the Reviewers and the Editors will be satisfied with the revisions for the manuscript. 
  Thanks for all the help.

Sincerely,

Comments:
-Section 1: Although references 7 and 8 are appropriate, I suggest including some older to show (as mentioned in the text) that there is an important body of evidence concerning this issue (for instance, cite the study from Ma et al., Ann Neurol 2014).
Response to comment: Thank you for the reviewer’s valuable comments. We have revised it and added the reference.

-Section 2, subheading 2.1: in the second paraghaph, please correct thee. In the same section, I strongly suggest to include a figure showing the structure and activation process of NLRP3, since this is one of the main points of the review.
Response to comment: Thank you for the reviewer’s valuable comments. We added the information, as shown in Figure 1.

-Section 2, subheading 2.2.: title. Do the authors believe that this section is focused on therapeutic targets for ICH or for secondary brain injury produced after ICH? Please modify according the statement (third paragraph) that reads as follows: Brain injury induced by inflammation after ICH can be alleviated by directly or indirectly inhibiting NLRP3 inflammasome activation.
Response to comment: Thank you for the reviewer’s valuable comments. We have revised it.

-In the same line, last paragraph of this section: the authors cite that: “Many drugs can improve brain dysfunction and prevent ICH injury by acting on NLRP3 inflammasome pathways”. However, the point is not to prevent the development of ICH but to improve the secondary brain injury… Please modify that sentence and also the heading of the table according to this.
Response to comment: Thank you for the reviewer’s comments. We have revised it.

-Fourth paragraph: authors use SBI abbreviation for "surgical brain injury" while it was previosly used for secondary brain injury. Please correct.
Response to comment: Thank you for the reviewer’s comments. We have revised it.

-Table 1: It should be very interesting and will strengthen the review if studies focused on humans are also included. Atorvastatin study (ref. 31) has also used human brains… It should be highlighted.
Response to comment: Thank you for the reviewer’s valuable comments. We have revised this sentence.

-Table 2: only three studies have been included. Please include more studies. I also strongly suggest changing the heading of the table. It should be better if it reads as: “The changes in gut microbiota after (instead of in) intracerebral haemorrhage”.
Response to comment: Thank you for the reviewer’s valuable comments. We have added the relevant data and information, but so far there are not many reports on it.

-Page 7: when talking about SCFAs (ref. 44) specify that is acetate the fatty acid responsible for the actions mentioned.
Response to comment: Thank you for the reviewer’s comments. We have revised it.

-Section 4  page 8: Please include the SBI abbreviation in the first paragraph
Response to comment: We have revised it.

-Figure 1 legend: the neuroinflammation plays (“s” added).
Response to comment: Thank you for your suggestion, we have revised it.

-Last paragraph of the manuscript: change “to change” by “by changing”.
Response to comment: Thank you for your suggestion, we have revised it.

Reviewer 2 Report

This review by Zhang et al is a very well-written, comprehensive manuscript covering a broad, yet focused, discussion on a highly relevant topic.

This reviewer has only a minor observation:

Please include citation(s) for the following statement in Section 2.2:

 "Various risk factors such as Ca2+ mobilization, Na+ influx, K+ efflux, chloride efflux, mitochondrial dysfunction, oxidative stress and lysosomal damage are involved in the activation of NLRP3 inflammasome to mediate neuroinflammatory responses after ICH."

Author Response

  Thank you for  the reviewer's comments concerning our manuscript. Those comments are all valuable and very helpful for revising and improving our paper. 
  We hope the Reviewers and the Editors will be satisfied with the revisions for the manuscript. 
  Thanks for all the help.

Sincerely,

Comments:
Please include citation(s) for the following statement in Section 2.2:
"Various risk factors such as Ca2+ mobilization, Na+ influx, K+ efflux, chloride efflux, mitochondrial dysfunction, oxidative stress and lysosomal damage are involved in the activation of NLRP3 inflammasome to mediate neuroinflammatory responses after ICH."
Response to comment: Thank you for the reviewer’s valuable comments. We have supplemented the relevant reference.